# Epidemiological Studies of Children’s Gut Microbiota: Validation of Sample Collection and Storage Methods and Microbiota Analysis of Toddlers’ Feces Collected from Diapers

**DOI:** 10.3390/nu14163315

**Published:** 2022-08-12

**Authors:** Hazuki Tamada, Yuki Ito, Takeshi Ebara, Sayaka Kato, Kayo Kaneko, Taro Matsuki, Mayumi Sugiura-Ogasawara, Shinji Saitoh, Michihiro Kamijima

**Affiliations:** 1Department of Occupational and Environmental Health, Graduate School of Medical Sciences, Nagoya City University, Mizuho-ku, Nagoya 4678601, Aichi, Japan; 2Department of Obstetrics and Gynecology, Graduate School of Medical Sciences, Nagoya City University, Mizuho-ku, Nagoya 4678601, Aichi, Japan; 3Department of Pediatrics and Neonatology, Graduate School of Medical Sciences, Nagoya City University, Mizuho-ku, Nagoya 4678601, Aichi, Japan

**Keywords:** gut microbiome, diaper, toddlers, epidemiology, feces

## Abstract

The composition of human gut microbiota influences human health and disease over the long term. Since the flora in specimens can easily change at ambient temperature outside the body, epidemiological studies need feasible methods of stool specimen collection and storage to be established. We aimed to validate two methods: feces frozen-stored in tubes containing guanidine thiocyanate solution for two months after collection (Method B), and feces excreted in diapers and frozen-stored (Method C). Validation was by comparison with a gold standard Method A. Bacterial flora of five adults were sampled and stored by all three methods. Bacterial composition was examined by amplicon sequencing analysis. Bland–Altman analyses showed that Methods B and C might change relative abundances of certain bacterial flora. Thereafter, we analyzed the bacterial flora of 76 toddlers (two age groups) in stools sampled and processed by Method C. The diversity indices of toddlers’ flora were less than those of adults. The relative abundance of some bacteria differed significantly between children aged 1.5 and 3 years. The specimen collection and storage methods validated in this study are worth adopting in large-scale epidemiological studies, especially for small children, provided the limited accuracy for some specific bacteria is understood.

## 1. Introduction

Microorganisms in the human intestinal tract number 10^14^ to 10^15^ cells, and have a symbiotic relationship with the host [1]. While the gut microbiota has beneficial functions for the host, such as digestion of polysaccharides, protection from pathogens, and support for the development of the immune system [2,3], dysbiosis has reportedly been associated with inflammatory bowel disease, irritable bowel syndrome, obesity, allergies, and psychiatric disorders [4]. In addition, evidence is accumulating that bacterial colonization affects host development and physiology, and health and disease over the long term [5,6]. Intestinal bacteria make up a complex “ecosystem” through in-depth interspecies interactions [7,8,9,10], and there are many reports at the genus and species levels of the physiological roles of microorganisms and their metabolites in humans. To clarify the relationship between human health and gut microbiota, it is necessary to clarify the composition of the gut microbiota in children that may affect their later health through epidemiological studies.

The storage conditions of specimens affect the results of bacterial flora analysis [11]. In general, the bacterial composition of feces changes after 15 min of storage at room temperature, which can be an obstacle to epidemiological studies; hence, it is desirable to immediately freeze fecal samples [12]. In previous studies of the gut microbiome of children, fecal samples were transported with cooling [13], in an anaerobic environment [14], using RNAlater [15,16], or fecal occult blood cards [17]. It is a heavy burden to both parents and researchers to collect samples from small children who cannot excrete autonomously. Our group previously established a method for extracting and analyzing urine samples collected from diapers [18]. If the use of diapers is also valid for sampling stools, it will be a very useful technique for investigating gut microbiota in epidemiological studies. A stool collection container that can be stored at room temperature was developed recently [19], although long-term storage of collected samples may be required in large-scale epidemiological studies from the viewpoint of cost and manpower. The present study aimed to achieve the following two goals: first, to validate methods that differ from the gold standard in terms of collection, storage temperature, and period; and secondly, to characterize the gut microbiota of toddler stool samples collected from diapers.

## 2. Materials and Methods

### 2.1. Study Design

This study comprised two parts, a method validation study (Research I), and a gut microbiota profiling study in toddlers (Research II). Research II was conducted as an Adjunct Study [20] of the Japan Environment and Children’s Study (JECS), a prospective observational birth cohort study that was outlined in the JECS protocol paper [21]. The ethics committee of the Nagoya City University Graduate School of Medical Sciences approved this study protocol (approval numbers 60-00-1116 and 70-19-0001).

#### 2.1.1. Research I: Gut Microbiota Analysis of Fecal Specimens Collected and Stored by Different Methods

Fecal samples were collected from five adult volunteers (#1–#5) by either the gold standard method (Method A) or two other methods that we consider to be more relevant to epidemiological studies (Methods B or C) and compared (Figure 1). In Method A, fecal samples were collected in collection tubes containing guanidine thiocyanate solution (TechnoSuruga Laboratory Co., Ltd., Shizuoka, Japan), kept under refrigeration, and shipped the next day to a contract laboratory (TechnoSuruga Laboratory) under refrigeration [19]. In Method B, fecal samples were collected in collection tubes in the same way as Method A, kept at room temperature for 3 h, stored at −80 °C for about two months until shipment, and then frozen-shipped. In Method C, urine and stool samples were applied to disposable diapers, kept at room temperature for 3 h, and subsequently stored under refrigeration for 24 h. After that, only feces were stored at −80 °C for about two months until shipment, and then frozen-shipped. Method C is a simulation from sampling to shipment, a plausible scenario in epidemiological studies targeting children with diapers.

#### 2.1.2. Research II: Analysis of Toddlers’ Feces Excreted in Diapers

Children participating in the Aichi regional subcohort of JECS, JECS-A [20], which comprised 43% of the children in that age group who were born in the study area, were recruited at the time of their 18-month checkup provided by the local government. Their guardians as legally acceptable representatives were asked to participate in the study, and their informed consent for the present study was obtained. The overall participation rate among recruited participants was 83.0%. The participants of the present study wore designated disposable diapers, which had been distributed in advance, during the night, and these were collected as refrigerated cargoes the next day after their use. When stools excreted in diapers were found, they were stored at −80 °C and frozen-shipped for analysis. Used diapers were collected from 2721 children between 22 June 2015 and 31 July 2016 (children were approximately 1.5 years old), and between 11 May 2016, and 20 December 2017 (children were approximately 3 years old). Of these, 76 fecal samples (from 26 boys and 29 girls at 16–21 months and from 15 boys and 6 girls at 35–39 months) were available, and all of them were included in this analysis. In addition, we obtained information from the questionnaire asked in the main study of the JECS on the mode of delivery, the feeding method during the first month after birth, and the starting date of feeding solid foods. The characteristics of the study participants are presented in Table 1.

### 2.2. Gut Microbiota Analysis

The DNA extraction and polymerase chain reaction were conducted according to a previously described method [22,23,24,25,26]. Identification of sequence reads was performed manually using the Ribosomal Database Project (RDP) Classifier tool ver 2.11, which is available from the RDP website (http://rdp.cme.msu.edu/classifier/, accessed on 1 March 2021) [27]. Bacterial species were identified from sequences using the Metagenome@KIN Version 2.2.1 analysis software (World Fusion Co., Ltd., Tokyo, Japan). The joined amplicon sequence reads were processed through QIIME2 (Quantitative Insights Into Microbial Ecology version 2) ver 2020.6 [28]. The quality filtering and chimeric sequences were deleted using DADA2 (Divisive Amplicon Denoising Algorithm 2) denoise-single plugin Version 2017.6.0 using the default settings [29]. Taxonomy was assigned using Greengenes database Version 13.8 by training a naive Bayes classifier [30]. In addition, the diversity indices (Chao1, Shannon, Simpson, observed operational taxonomic units (OTUs), and Faith’s phylogenetic diversity (PD)) were calculated.

These analyses of fecal microbiota were outsourced to TechnoSuruga Laboratory Co., Ltd., Shizuoka, Japan.

### 2.3. Statistical Analysis

In Research I, Bland–Altman analyses were performed in each of the three relative abundance (%) ranges (“≥10”, “≥1 and <10”, and “<1”) at phylum and class levels and in four abundance ranges (“≥10”, “≥1 and <10”, “≥0.1 and <1”, and “<0.1”) at the order, family, genus, and species levels. Limits of agreement (LOAs, mean ± 1.96 × standard deviation (SD) of differences between two measurements) and 95% confidence intervals (CIs) of their upper/lower limits were calculated and plotted. In addition, the Firmicutes/Bacteroidetes (F/B) ratio was calculated.

Comparisons were made between Method A, the gold standard [19], and Method B or C. Dunnett’s test was performed to compare the diversity indices of Methods B and C with those of Method A as a control. Kruskal–Wallis and Mann–Whitney U tests were performed, and effect sizes were calculated to compare the F/B ratios. In these tests, *p* values were corrected by the Bonferroni method. Heatmap and cluster analyses (Ward’s method) were performed for log-converted relative abundance (%).

In Research II, Dunnett’s test was applied to compare the diversity indices between the adults and both children (1.5 years and 3 years) groups. Mann–Whitney U test was performed to compare the relative abundance between the two age groups of the children (1.5 years and 3 years). All analyses were performed using R version 3.6.2.

In the present study, a *p* value of <0.05 was considered statistically significant.

## 3. Results

### 3.1. Research I: Gut Bacterial Compositions According to Three Different Methods

The bacterial compositions of samples from each subject are shown in Figure 2 and Appendix A (see Appendix A for IDs of bacteria depicted in the figures). At the phylum level, Firmicutes (P1), Bacteroidetes (P2), Actinobacteria (P3), and Proteobacteria (P4) accounted for more than 96% of the occupancy in all the methods (Figure 2a). Method B yielded microbes at the phylum, class, order, family, genus, and species levels which did not differ markedly from Method A in their occupancy (Figure 2 and Appendix A). In Method C, at the phylum level, the relative abundance of Bacteroidetes (P2) tended to decrease and Actinobacteria (P3) tended to increase (Figure 2a). The decrease of P2 relative abundance was attributable to bacteria that belonged to Bacteroidia (C2) at the class level (Appendix A), Bacteroidales (O2) at the order level (Appendix A), Bacteroidaceae (F2) at the family level (Appendix A), and *Bacteroides* (G1) at the genus level (Figure 2b). The increased Actinobacteria (P3) at the class, order, family, and genus levels mainly belonged to Actinobacteria (C3) (Appendix A), Bifidobacteriales (O3) (Appendix A), Bifidobacteriaceae (F4) (Appendix A), and *Bifidobacterium* (G2) (Figure 2b), and Coriobacteriia (C5) (Appendix A), Coriobacteriales (O5) (Appendix A), Coriobacteriaceae (F7) (Appendix A), and *Collinsela* (G6), respectively (Figure 2b).

The Chao1, Shannon, and Simpson diversity indices, the observed OTUs, and Faith’s PD did not differ significantly in Methods B and C compared to Method A (Figure 3a). The F/B ratio in Method C differed but not significantly compared with Method A (*p* = 0.188, Figure 3b), although the effect size was 0.589.

Bland–Altman plots in each relative abundance range category at the phylum, class, order, family, genus, and species levels (Figure 4, Appendix A, Figure 5 and Appendix A, respectively) revealed that LOAs and 95% CIs of their upper/lower limits in the comparison of Methods A–C were much wider than those in the comparison of Methods A–B, which indicated wider random fluctuations around the means in the Methods A–C comparison. In general, LOAs in both analyses covered 0, which indicated that systemic errors were not evident.

Table 2 shows that at the phylum level, the detection frequencies of bacteria at 1% or higher as the maximum relative abundance in any of the specimens were the same between Methods A and B (Table 2a), and between Methods A and C (Table 2b). Firmicutes (P1), Bacteroidetes (P2), Actinobacteria (P3), Proteobacteria (P4), and Verrucomicrobia (P6) fell within the 1% or higher category, while Fusobacteria (P5) did not, in the Methods A–C comparison (Table 2b), because its detection at ≥1% occurred only in Method B. Table 2 also shows that part of the abundance differences between the methods exceeded LOAs. For example, Bacteroidetes (P2) was detected in all five subjects in Methods A, B, and C, and the relative abundance was more than 10% in at least one specimen in each of the method pairs. When looking at the abundance differences in each specimen for this abundance range category, the Methods A–B comparison showed that the difference of the bacterium (P2) in one specimen (#1) was outside the LOA (Figure 4a, left panel), while there were no specimens in which the differences were outside the LOA in the Methods A–C comparison (Figure 4a, right panel). When taking the 95% CIs for the LOA into account, the differences in two specimens (#1 and #4) were outside an upper limit of 95% CI of a lower limit of LOA in the Methods A–B comparison (Figure 4a, left panel) and that in three specimens (#1, #4, and #5) were outside a lower limit of 95% CI of an upper limit of LOA in the Methods A–C comparison (Figure 4a, right panel). As for phyla that were detected at the maximum relative abundance of <1%, the frequencies of detection of the relevant phylum were not the same between the methods (Table 2).

The same analyses at the class, order, family, genus, and species levels are shown in Appendix A, Table 3 and Appendix A, respectively. Again, the detection frequencies of bacteria whose relative abundance was >1% were almost the same among the methods, although part of the abundance differences between the methods exceeded LOAs (Appendix A, Figure 5 and Appendix A). As for the bacteria whose maximum abundances were ≤1%, the differences in their detection frequencies and relative abundances between the methods varied depending on the bacteria. For example, *Clostridium* (G42) showed stable detection (5/5) in all the methods with the maximum abundance of ≥0.1 and <1%, and the abundance differences between the Methods A–B pairs were within the LOAs (Table 3a). However, for some other bacteria, the detections were not so stable. When bacteria were detected by only one of the methods, the Bland–Altman plots were fan-shaped especially in the <0.1% range categories (Figure 5d and other Appendix A).

Heatmap and cluster analyses at the genus and species levels showed that bacterial compositions identified by the three methods in each specimen collected from the same subject were classified into the same clusters that corresponded to each subject (#1–#5) (Figure 6 and Appendix A), but at the higher-level taxonomic categories, the clusters did not correspond to the subjects (Appendix A).

### 3.2. Research II: Bacterial Composition of Toddlers’ Feces Excreted in Diapers

Bacterial composition, at the phylum and genus levels, of 76 toddlers’ feces excreted in diapers is shown in Figure 7. There was no clear trend in the composition of the predominant bacteria at the phylum and genus levels according to age or mode of delivery.

Diversity indices of the toddlers’ specimens were significantly lower than those of adults obtained by Method C in Research I, except for the Simpson diversity index (Table 4).

Table 5 summarizes the results of the comparison between the 1.5 years and 3 years age groups for bacteria, for which Method C was validated in Research I, that were reported as potentially beneficial or detrimental to health in previous studies [7,8,31,32,33] and their coefficients of variation in each age group. For example, there was a significant difference between the 1.5 years and 3 years age groups in the relative abundance of *Blautia* (G5, *p* < 0.001). Figure 8 shows the bacteria whose abundance differences in the Methods A–C comparison were within the LOA, with significant differences between the 1.5 years and 3 years age groups in their relative abundances. *Lactobacillus* (G56) had a larger variation (relatively larger coefficients of variation of 1.980 and 3.000, respectively).

## 4. Discussion

The primary conclusion from Research I is that both Method B and C, the latter of which is more relevant to epidemiological studies targeting children with diapers, are worth adopting when Method A, a gold standard method, is difficult to adopt. Based on this finding, Research II was conducted, and it revealed that the bacterial flora composition of feces of toddlers differed from that of adults, as reported in previous studies [14]. Furthermore, the relative abundances of some bacteria differed between the ages of 1.5 and 3 years.

When adopting Method B or Method C, it is important to recognize the following limitation of the methods: bacterial flora composition resulting from these methods could change depending on their taxonomic category, (phylum/class/order/family/genus/species), when compared with Method A. Relative abundance differences of some bacteria were outside the LOA in Bland–Altman analyses; therefore, care should be taken when implementing Methods B and C and interpreting their results. Especially in Method C, the F/B ratio, which has been linked to obesity (although a complete consensus has not been reached) [34,35], deviated markedly; therefore, it is not appropriate to use this measure in diaper stools. However, the bacterial compositions as a whole at the genus and species levels identified by these methods had enough resolution to differentiate individual subjects. Diversity indices did not differ significantly among the methods, which suggested that bacterial community heterogeneities were maintained in Methods B and C.

The differences in bacterial composition in samples collected by Methods B or C compared to Method A may be due to different conditions during collection, storage, and transportation. Bacterial growth is influenced by several factors in the environment [36]. For example, the relative abundance of *Bacteroides* (G1) tended to decrease in Method C, which might have been because the bacteria are obligate anaerobes and their abundance was possibly affected by the aerobic environment of the diaper. However, the variation of each occupancy in the present study cannot be explained by oxygen demand alone, and must be examined in more detail in the future. The LOAs of the Bland–Altman analysis of Methods A–C were much larger than those of Methods A–B, and the 95% CIs of their upper and lower limits were also wider, probably due to wider random fluctuations of the measurement in Method C, which led to a more strict agreement range in Method B than in Method C. Method B was thus considered more sensitive than Method C for detecting statistically significant differences among the bacteria that could be tested by the methods. The bias of each measurement in both methods was not unidirectional in any taxonomic category, suggesting that the bias was due to the characteristics of each bacterial taxon.

Bacterial flora analysis of feces collected from toddlers’ diapers in Research II showed that their bacterial composition differed from that of adults, which replicated previous findings showing lower diversity compared to adults [14]. In addition, the abundance of bacteria that have been reported to be beneficial or detrimental in previous studies [7,8,31,32,33] was compared between children aged 1.5 years and 3 years. For example, the abundance of *Blautia* (G5), which is considered beneficial because of its significantly lower occupancy in older people with obesity [7,31], differed significantly between children aged 1.5 and 3 years. Such a comparison was possible since the abundance difference between Methods A and C was within the LOA in Research I. Another example in which the comparison was also possible was *Bacteroides fragilis* (S73), which was reportedly opportunistically harmful but also regarded as a beneficial species that might inhibit inflammation through the production of polysaccharide A [7,8,33]. No significant difference in abundance was observed between the two ages (Table 5).

The gut microbiota reportedly undergoes drastic changes in the first few years of life, including the increased diversity within an individual, a decrease in lactic acid bacteria, and increases in other bacteria such as Bacteroidaceae and Lachnospiraceae [14,37,38]. In an gut microbiota analysis study of 513 children and adults, specimens were stored at −80 °C within 30 min of their collection [39]. Another toddler gut microbiota study used commercial sampling tubes [40]. Gut microbiota analysis of toddlers using disposable diapers has also been conducted, but its validity has not been addressed [41,42]. The method of immersing stool specimens in ethanol is reportedly useful in areas such as Africa where resources and facilities may not be available [43]. Although there have been reports on the gut microbiota of toddlers, the human gut microbiota is influenced by factors such as race, genetics, country and region [6], and should be studied in various regions under different environments.

Microbiome research is currently actively conducted worldwide, and the function of various bacteria is likely to be further clarified in the future. Research II showed relatively higher coefficients of variation in some bacteria, which indicated relatively larger differences in their abundance among individuals, suggesting the possibility that these bacteria might play some role in the different predispositions for health or disease. Furthermore, the results of this study may be applicable not only to infants and toddlers, but also to the elderly who use diapers for elimination. The findings of this study will contribute to the further study of intestinal microbiomes, especially within epidemiological studies.

A limitation of this study is that in Research I, we could not examine the method validity for bacteria that were detected only in children because we used adult specimens. We used adult specimens for the following reason: the fecal bacterial composition of adults is more stable and more complex than that of toddlers, and individual differences are smaller [44], and we contemplated that comparisons between the methods were possible with the small number of adult volunteer specimens. Further studies, including those of bacteria that were not detected in the present study, are desirable in the future. Another limitation of this study is that in Research II, the comparison between ages was not based on the same participants, and this might have affected the results.

## 5. Conclusions

Both Method B and C are worth adopting when Method A, a gold standard method, is difficult to adopt, especially in large-scale epidemiological studies. When the study population is young children who cannot excrete autonomously, Method C is a practical strategy for investigating certain fecal bacteria of which the difference in abundance is within the LOA. In such studies, the interpretation of the data should be made with caution for bacteria of which the accuracy is reduced. Especially, the researchers should not use the results of bacteria for which the agreement between Methods A and C is poor. Even for bacteria whose abundance differences between Methods A and C were within the limits of agreement in all the specimens in Research I, larger sample sizes are desirable because random fluctuations around the means in the Methods A–C comparison were wider than those in the Methods A–B comparison. However, sending diapers back to the research team seems to be a relatively low hurdle for parents of the study participants (the diaper return rate was 88.4% in our previous study [18]). Thus, this method is acceptable to participants and has the advantage of reducing selection bias in epidemiological studies. As for profiles of the gut microbiota of toddlers, the observations in this study confirmed that their flora composition was different from that of adults, and that the relative abundance of some bacteria differed between the ages of 1.5 and 3 years.

## Figures and Tables

**Figure 1 nutrients-14-03315-f001:**
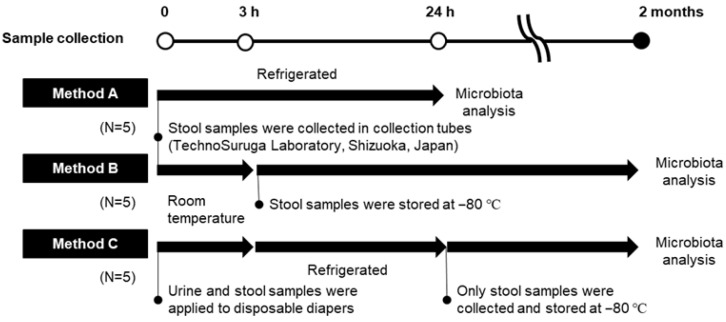
Experimental overview of Research I. For Methods A and B, stool samples were collected using commercial collection tubes. For Method C, urine and stool samples were applied to disposable diapers. For Methods B and C, the samples were stored at −80 °C for 2 months.

**Figure 2 nutrients-14-03315-f002:**
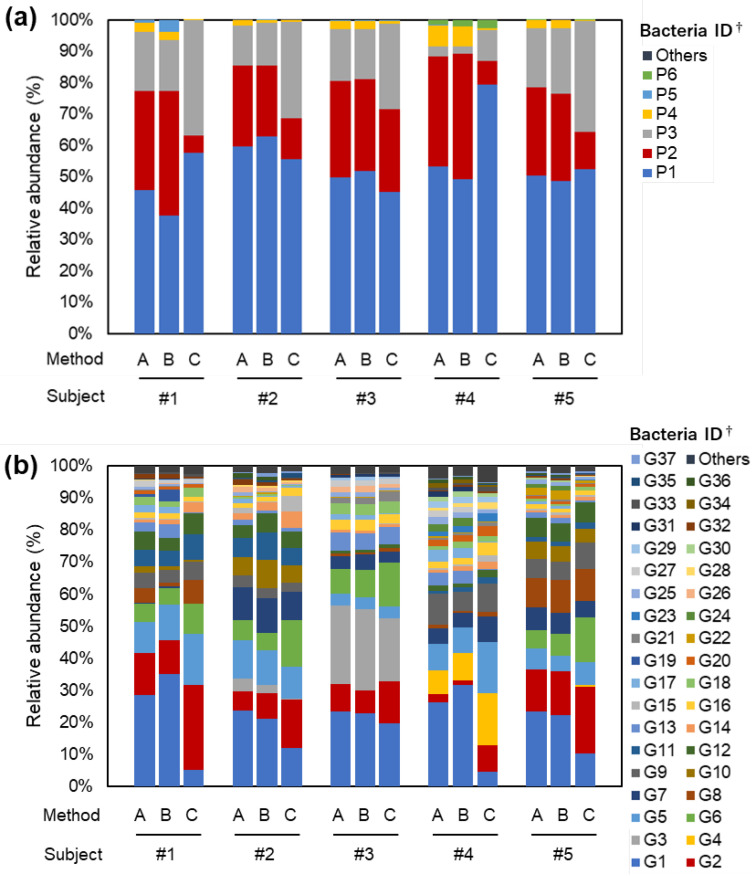
Research I: Fecal bacterial composition in five adult volunteers. Bacterial composition at the (**a**) phylum and (**b**) genus levels. Bacteria detected at a relative abundance of less than 1% were grouped together as Others. ^†^ See Appendix A or Appendix A for the bacterial taxonomy name corresponding to each ID.

**Figure 3 nutrients-14-03315-f003:**
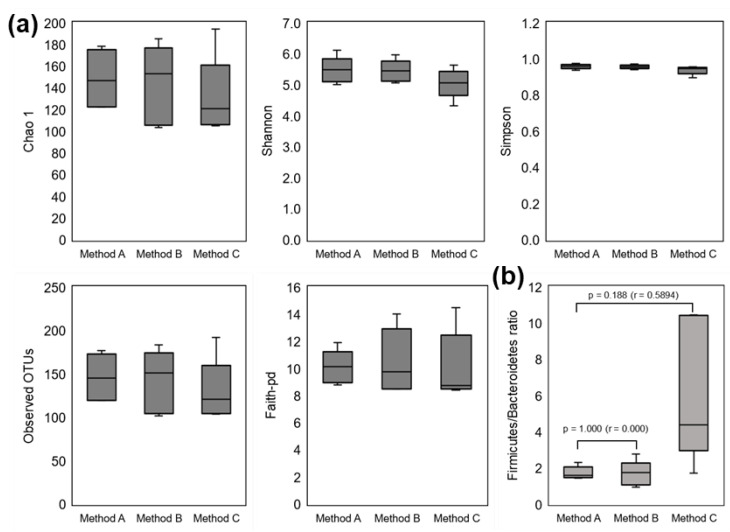
Research I: Diversity indices and the Firmicutes/Bacteroidetes ratio of the fecal bacterial community in five adult volunteers. (**a**) Diversity indices and the (**b**) Firmicutes/Bacteroidetes ratio in each method.

**Figure 4 nutrients-14-03315-f004:**
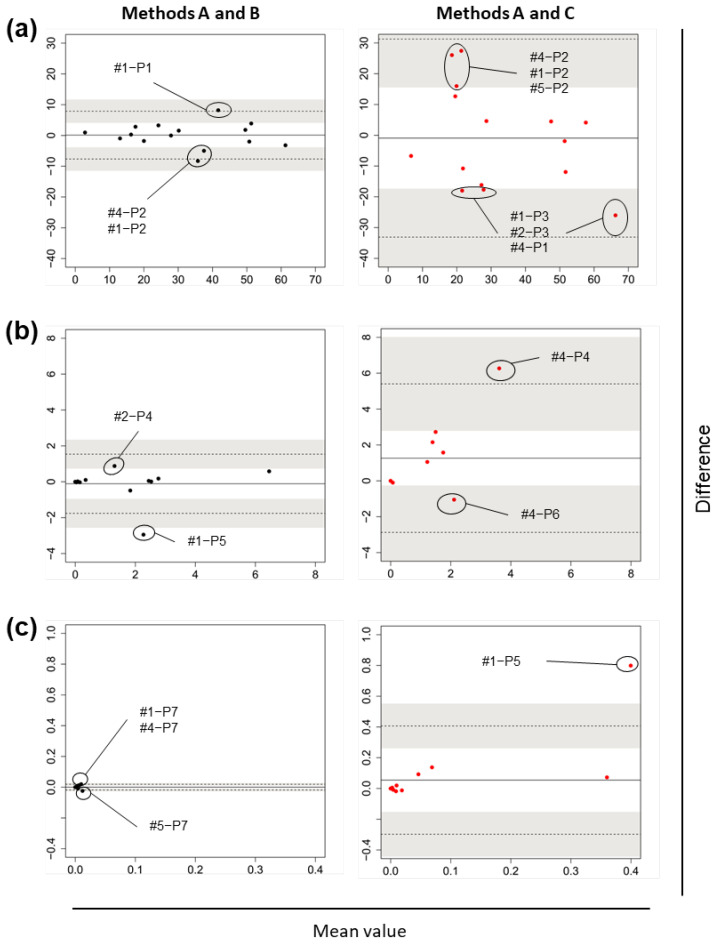
Research I: Bland–Altman plots of the relative abundance of fecal bacteria at the phylum level obtained by each method from five adult volunteers. Comparisons between Methods A–B (left) and Methods A–C (right). The relative abundance of bacteria in each specimen was plotted. (**a**) Their maximum abundance in any of the specimens was ≥10%; (**b**) ≥1% and <10%; (**c**) <1%. Solid lines indicate the means of the differences between the two test values, dotted lines indicate the limits of agreement (the mean of the difference ± 1.96 × standard deviation), and the gray shaded area indicates the 95% confidence interval of the limits of agreement. For Methods A and B, stool samples were collected using commercial collection tubes. For Method C, urine and stool samples were applied to disposable diapers. For Methods B and C, the samples were stored at −80 °C for 2 months.

**Figure 5 nutrients-14-03315-f005:**
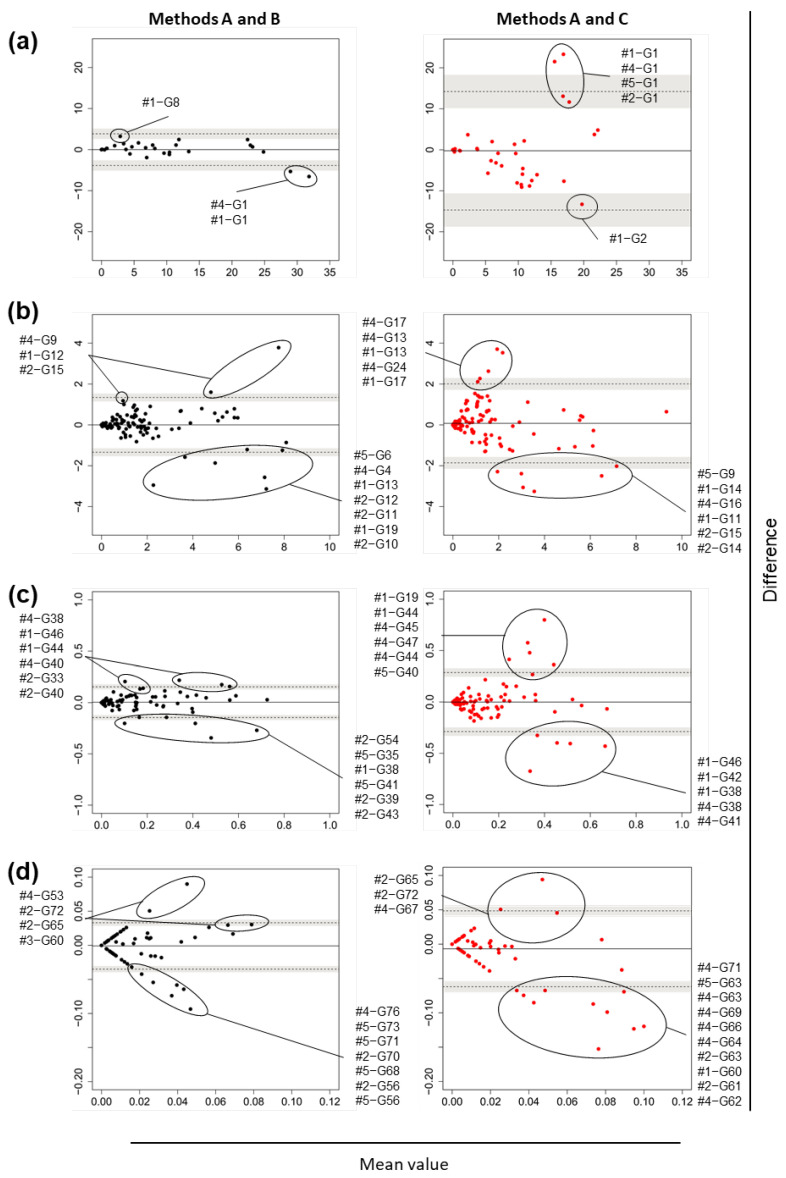
Research I: Bland–Altman plots of the relative abundance of fecal bacteria at the genus level obtained by each method from five adult volunteers. Comparisons between Methods A–B (left) and Methods A–C (right). The relative abundance of bacteria in each specimen was plotted. (**a**) Their maximum abundance in any of the specimens was ≥10%; (**b**) ≥1% and <10%; (**c**) ≥0.1% and <1%; (**d**) <0.1%. Solid lines indicate the means of the differences between the two test values, dotted lines indicate the limits of agreement (the mean of the difference ± 1.96 × standard deviation), and the gray shaded area indicates the 95% confidence interval of the limits of agreement. For Methods A and B, stool samples were collected using commercial collection tubes. For Method C, urine and stool samples were applied to disposable diapers. For Methods B and C, the samples were stored at −80 °C for 2 months.

**Figure 6 nutrients-14-03315-f006:**
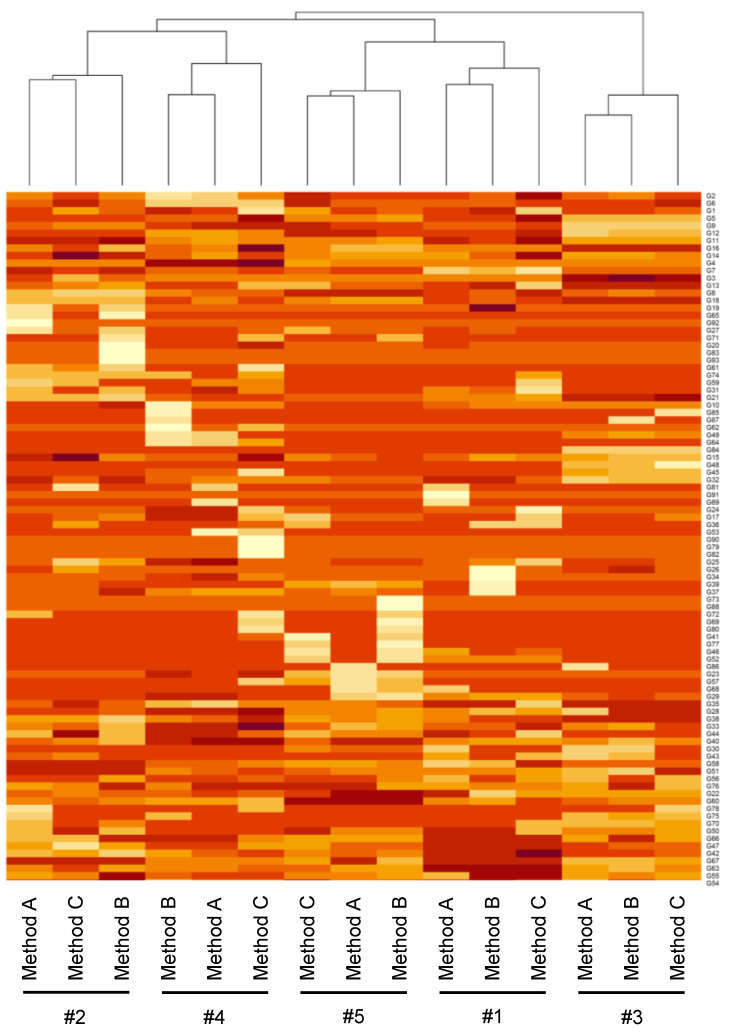
Research I: Heatmap of the gut microbiota composition at the genus level in five adult volunteers. Heatmap of log-transformed values of the relative abundance at the genus level in each method of Research I, using Ward’s method for clustering.

**Figure 7 nutrients-14-03315-f007:**
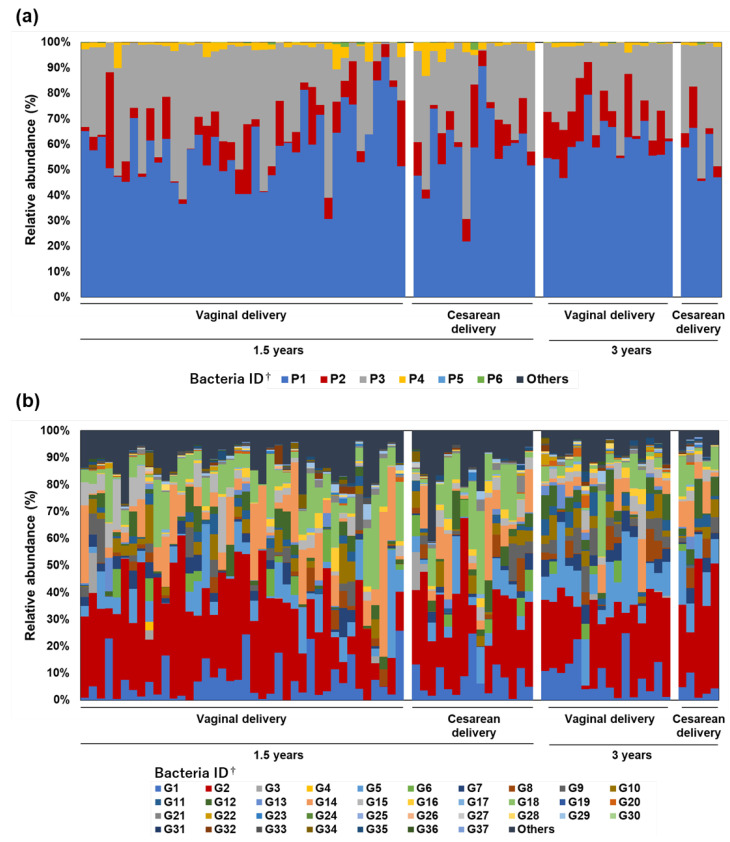
Research II: Fecal bacterial composition at the phylum and genus levels in 76 toddlers. (**a**) Phylum and (**b**) genus levels. Bacteria detected at less than 1% of their relative abundance were grouped together as Others. These were sorted by age and mode of delivery. ^†^ See Appendix A or Appendix A for the bacterial taxonomy name corresponding to each ID.

**Figure 8 nutrients-14-03315-f008:**
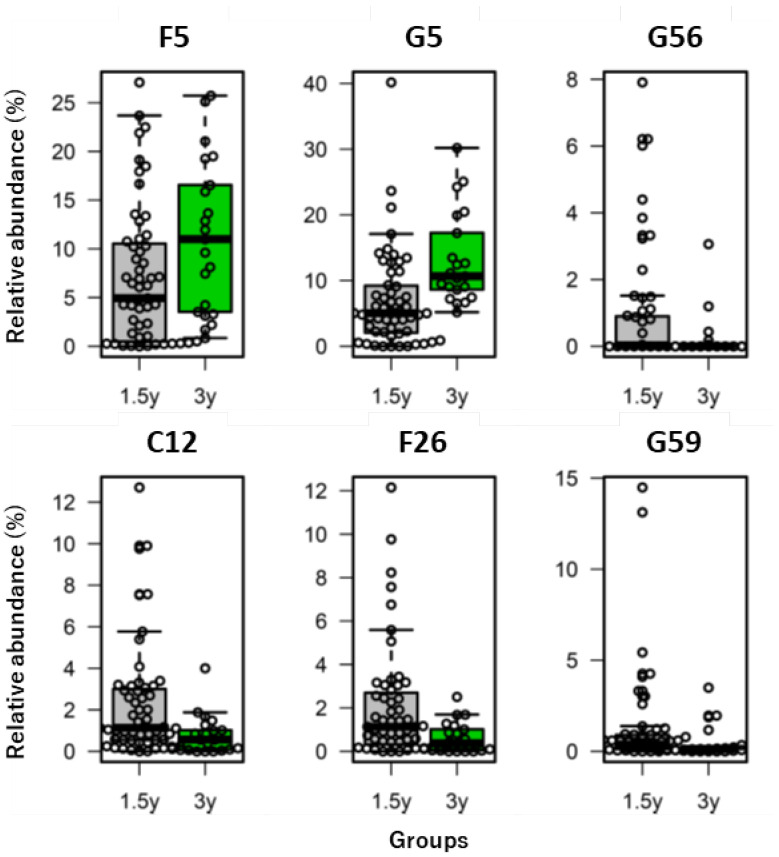
Box plots of the bacteria listed in Table 5 that showed significant different relative abundances between the two age groups (1.5 years and 3 years). The depicted bacteria are those whose abundance differences between Methods A–C were within the limits of agreement in all the specimens (See Table 5). Bold lines in the middle of the box are the medians, the top of the box is the 3rd quartile, the bottom of the box is the 1st quartile, the upper whiskers are the largest data less than or equal to “3rd quartile + 1.5 × (3rd quartile − 1st quartile)” and the lower whiskers are the smallest data equal to “1st quartile − 1.5 × (3rd quartile − 1st quartile)”. See Appendix A for the bacterial taxonomy name corresponding to each ID.

**Table 1 nutrients-14-03315-t001:** Characteristics of the study participants in Research II.

		Total	Age during Stool Collection			
			1.5 Years		3 Years	
			n	(%)	n	(%)
Sex						
	Male	41	26	(47)	15	(71)
	Female	35	29	(53)	6	(29)
Mode of delivery						
	Spontaneous delivery	33	23	(42)	10	(48)
	Induced delivery	18	13	(24)	5	(24)
	Vacuum extraction	5	4	(7)	1	(5)
	Planned Cesarean delivery	19	14	(25)	5	(24)
	/Emergent Cesarean delivery					
	Missing	1	1	(2)	0	(0)
Feeding method during the first month after birth						
	Breastfeeding only	36	27	(49)	9	(43)
	Mixed feeding	37	25	(45)	12	(57)
	Infant formula only	2	2	(4)	0	(0)
	Missing	1	1	(2)	0	(0)
Starting date of feeding solid foods						
	4 months old	1	1	(2)	0	(0)
	5 months old	38	26	(47)	12	(57)
	6 months old	29	22	(40)	7	(33)
	7 months old	3	2	(4)	1	(5)
	8 months old	1	1	(2)	0	(0)
	Missing	4	3	(5)	1	(5)

**Table 2 nutrients-14-03315-t002:** Research I: Summary of bacteria detection and Bland–Altman plotting results (Methods A–B (**a**) and Methods A–C (**b**) at the phylum level.

(a)					
Maximum Relative Abundance	Bacteria ID ^†^	Number of Specimens with the Bacteria Detection (/5)		Number of Specimens with the Bacteria Abundance Out of the LOA	
		Method A	Method B	SD ^€^	CI ^∫^
≥10%	P1	5	5	1	1
	P2	5	5	1	2
	P3	5	5	0	0
≥1%, <10%	P4	5	5	0	1
	P5	4	4	1	1
	P6	2	2	0	0
<1%	P7	2	1	2	3
	P9	0	1	0	0
	P10	1	0	0	0
**(b)**					
**Maximum Relative** **Abundance**	**Bacteria ID ^†^**	**Number of Specimens with the Bacteria Detection (/5)**		**Number of Specimens with the Bacteria Abundance Out of the LOA**	
		**Method A**	**Method C**	**SD ^€^**	**CI ^∫^ **
≥10%	P1	5	5	0	1
	P2	5	5	0	3
	P3	5	5	0	2
≥1%, <10%	P4	5	5	1	1
	P6	2	2	0	1
<1%	P5	4	1	1	1
	P7	2	2	0	0
	P8	0	1	0	0
	P10	1	0	0	0

^†^ See Appendix A for the bacterial taxonomy name corresponding to each ID. ^€^ SD, LOA was defined as mean ± 1.96 × SD of the differences between two measurements. ^∫^ CI, LOA was defined as the lower or upper limit of 95% CI of the upper or lower limit, respectively, of SD. LOA, limit of agreement; SD, standard deviation; CI, confidence interval.

**Table 3 nutrients-14-03315-t003:** Research I: Summary of bacteria detection and Bland–Altman plotting results (Methods A–B (**a**) and Methods A–C (**b**)) at the genus level.

(a)					
Maximum Relative Abundance	Bacteria ID ^†^	Number of Specimens with the Bacteria Detection (/5)		Number of Specimens with the Bacteria Abundance Out of the LOA	
		Method A	Method B	SD ^€^	CI ^∫^
≥10%	G1	5	5	2	2
	G2	5	5	0	0
	G3	2	2	0	0
	G5	5	5	0	0
	G7	5	5	0	0
	G8	5	5	0	1
≥1%, <10%	G4	1	1	0	1
	G6	4	4	0	1
	G9	4	4	1	1
	G10	3	4	1	1
	G11	4	4	1	1
	G12	5	5	2	2
	G13	5	5	1	1
	G14	5	5	0	0
	G15	5	5	0	1
	G16	5	5	0	0
	G17	5	5	0	0
	G18	5	5	0	0
	G19	2	2	1	1
	G20	3	4	0	0
	G21	5	5	0	0
	G22	4	5	0	0
	G23	2	2	0	0
	G24	2	2	0	0
	G25	5	5	0	0
	G26	2	3	0	0
	G27	4	4	0	0
	G28	5	4	0	0
	G29	4	4	0	0
	G30	4	3	0	0
	G31	4	4	0	0
	G32	5	5	0	0
	G34	1	2	0	0
	G36	3	4	0	0
	G37	3	4	0	0
≥0.1%, <1%	G33	5	5	0	1
	G35	3	3	0	1
	G38	4	4	2	2
	G39	2	3	1	1
	G40	5	5	1	2
	G41	0	1	1	1
	G42	5	5	0	0
	G43	3	2	1	1
	G44	5	5	1	1
	G45	2	2	0	0
	G46	1	1	1	1
	G47	4	4	0	0
	G48	1	1	0	0
	G49	2	2	0	0
	G50	4	4	0	0
	G51	4	4	0	0
	G52	1	2	0	0
	G54	5	5	0	1
	G55	5	3	0	0
	G57	1	1	0	0
	G58	4	4	0	0
	G59	2	0	0	0
<0.1%	G53	1	0	1	1
	G56	2	2	2	2
	G60	4	4	0	1
	G61	1	1	0	0
	G62	0	1	0	0
	G63	4	4	0	0
	G64	1	1	0	0
	G65	1	1	0	1
	G66	3	1	0	0
	G67	2	3	0	0
	G68	2	1	1	1
	G69	0	1	0	0
	G70	2	2	1	1
	G71	0	2	1	1
	G72	1	1	1	1
	G73	0	1	1	1
	G74	1	2	0	0
	G75	2	2	0	0
	G76	3	3	0	1
	G77	0	1	0	0
	G78	1	0	0	0
	G80	0	1	0	0
	G81	2	0	0	0
	G83	0	1	0	0
	G84	1	1	0	0
	G85	0	1	0	0
	G86	2	0	0	0
	G87	0	2	0	0
	G88	0	1	0	0
	G89	2	0	0	0
	G91	1	0	0	0
	G92	1	0	0	0
	G93	0	1	0	0
**(b)**					
**Maximum Relative** **Abundance**	**Bacteria ID ^†^**	**Number of Specimens with the Bacteria Detection (/5)**		**Number of Specimens with the Bacteria Abundance Out of the LOA**	
		**Method A**	**Method C**	**SD ^€^**	**CI ^∫^**
≥10%	G1	5	5	2	4
	G2	5	5	0	1
	G3	2	2	0	0
	G4	1	2	0	0
	G5	5	5	0	0
	G6	4	4	0	0
	G7	5	5	0	0
	G8	5	5	0	0
≥1%, <10%	G9	4	4	1	1
	G10	3	3	0	0
	G11	4	4	1	1
	G12	5	5	0	0
	G13	5	5	2	2
	G14	5	5	2	2
	G15	5	5	1	1
	G16	5	5	1	1
	G17	5	5	2	2
	G18	5	5	0	0
	G20	3	3	0	0
	G21	5	5	0	0
	G22	4	5	0	0
	G23	2	2	0	0
	G24	2	2	1	1
	G25	5	5	0	0
	G26	2	2	0	0
	G27	4	4	0	0
	G28	5	4	0	0
	G29	4	3	0	0
	G30	4	2	0	0
	G31	4	3	0	0
	G32	5	5	0	0
	G33	5	5	0	0
	G34	1	1	0	0
	G35	3	4	0	0
	G36	3	4	0	0
≥0.1%, <1%	G19	2	0	1	1
	G37	3	3	0	0
	G38	4	4	2	2
	G39	2	2	0	0
	G40	5	5	0	1
	G41	0	2	1	1
	G42	5	5	1	1
	G43	3	2	0	0
	G44	5	4	2	2
	G45	2	2	1	1
	G46	1	2	1	1
	G47	4	4	1	1
	G48	1	1	0	0
	G49	2	1	0	0
	G50	4	3	0	0
	G51	4	3	0	0
	G52	1	2	0	0
	G53	1	1	0	0
	G54	5	5	0	0
	G55	5	4	0	0
	G56	2	3	0	0
	G57	1	2	0	0
	G58	4	3	0	0
	G59	2	3	0	0
<0.1%	G60	4	4	1	1
	G61	1	2	1	1
	G62	0	1	1	1
	G63	4	4	3	3
	G64	1	1	1	1
	G65	1	0	1	1
	G66	3	4	1	1
	G67	2	3	0	1
	G68	2	0	0	0
	G69	0	1	1	1
	G70	2	2	0	0
	G71	0	1	1	1
	G72	1	1	1	1
	G74	1	3	0	0
	G75	2	1	0	0
	G76	3	3	0	0
	G77	0	1	0	0
	G78	1	2	0	0
	G79	0	1	0	0
	G80	0	1	0	0
	G81	2	1	0	0
	G82	0	1	0	0
	G84	1	1	0	0
	G85	0	1	0	0
	G86	2	0	0	0
	G89	2	0	0	0
	G90	0	1	0	0
	G91	1	0	0	0
	G92	1	0	0	0

^†^ See Appendix A for the bacterial taxonomy name corresponding to each ID. ^€^ SD, LOA was defined as mean ± 1.96 × SD of the differences between two measurements. ^∫^ CI, LOA was defined as the lower or upper limit of 95% CI of the upper or lower limit, respectively, of SD. LOA, limit of agreement; SD, standard deviation; CI, confidence interval.

**Table 4 nutrients-14-03315-t004:** Diversity indices of the adult and toddler gut microbiota.

	Adults (Research I, Method C)	1.5 Years Group (Research II)	3 Years Group (Research II)
Chao1	131.2 ± 36.5	64.9 ± 12.8 **	83.9 ± 16.4 **
Shannon	5.0 ± 0.5	4.4 ± 0.5 *	4.7 ± 0.5
Simpson	0.94 ± 0.02	0.91 ± 0.04	0.92 ± 0.03
Observed operational taxonomic units	129.4 ± 36.4	64.8 ± 12.8 **	83.8 ± 16.4 **
Faith’s phylogenetic diversity	10.1 ± 2.6	5.9 ± 1.0 **	7.0 ± 1.2 **

* *p* < 0.001, ** *p* < 0.0001 when compared to adults by Dunnett’s test.

**Table 5 nutrients-14-03315-t005:** Between-age comparisons of potentially beneficial or detrimental bacteria detected in Research II.

Functionality	References	Category	ID ^†^	*p*-Value ^¶^	Coefficient of Variation ^§^	
					1.5 Years Group	3 Years Group
Beneficial	[31]	Family	F5	0.018	1.011	0.697
	[7,31]	Genus	G5	0.000	1.051	0.529
	[7,32]		G10	0.194	1.235	0.862
	[31,32]		G12	0.077	1.159	0.847
	[7]		G23	0.331	3.120	1.844
	[31]		G25	0.071	1.267	1.234
	[7]		G56	0.019	1.980	3.000
	[32]		G57	0.551	3.839	4.583
	[7,8]	Species	S29	0.331	3.120	1.844
	[32]		S40	0.079	2.266	1.384
	[7]		S43	0.917	0.774	0.701
	[7]		S127	0.286	5.204	-
Both	[7,8,33]	Species	S73	0.896	1.993	2.452
Detrimental	[8]	Class	C12	0.003	1.209	1.242
	[31]	Family	F7	0.515	1.352	1.333
	[8]		F26	0.003	1.274	1.157
	[31]	Genus	G6	0.260	1.814	2.023
	[7,32]		G59	0.015	2.178	1.869
	[7]	Species	S147	0.476	7.416	4.583
	[7]		S167	0.139	7.416	4.583

Bacteria presented in this table are those of which all the Bland–Altman plots were within the lower or upper limit of the 95% CI of the upper or lower limit, respectively, of SD as the LOA in Research I. See footnotes of Table 2 or Table 3 for definition of SD and CI. LOA, limit of agreement; SD, standard deviation; CI, confidence interval. -, bacteria not detected. ^†^ See Appendix A for the bacterial taxonomy name corresponding to each ID. ^¶^ Mann–Whitney U test for a between-age comparison of the bacterial relative abundance. ^§^ Calculated as standard deviation/mean.

## Data Availability

Data are available upon reasonable request.

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
