# Peer review of "Epidemiological Studies of Children’s Gut Microbiota: Validation of Sample Collection and Storage Methods and Microbiota Analysis of Toddlers’ Feces Collected from Diapers"

_nutrients, 2022, doi:10.3390/nu14163315_

Round 1

Reviewer 1 Report

It is interesting that the authors compared microbial composition in stool samples collected from 5 adults that were stored in different methods. It seems from the analysis that microbial composition between method A and method B (stool samples were stored in -80 after 3 hours of collection) do not differ much. However, method C differs from methods A and B. It is understandable that method C might be a wider choice of sample collection and storing if it gives comparable results with the gold standard. However, it requires further validation to adopt that method C is comparable to gold standard method A. Authors should analyze the microbiota of toddlers with method A as well and compare it with the method C that they already used. That will help to understand how correctly method C can be adopted for future studies.

Author Response

Dear Reviewer 1

We sincerely appreciate your positive review. We agree with your point that further validation is required to adopt Method C as a comparable method as the gold standard Method A. However, our point is not to conclude that Method C is comparable as Method A, but to provide information about which bacteria is affected when using the Method C as a practical approach to reduce a selection bias in epidemiological studies. In such studies, the researchers should understand that the accuracy is reduced for some specific bacteria that are plotted outside the limits of agreement in Figures 4, 5, S1, S2, S3, S4, and S5, of which the taxonomic names are listed in Tables S1-S6. Therefore, the readers of this article can recognize the accuracies of the presented data collected from toddlers using Method C in Research â…¡, and can interpretate the results. This point is phrased as ‘The specimen collection and storage methods validated in this study are worth adopting in large-scale epidemiological studies, especially for small children, provided the limited accuracy for some specific bacteria is understood.’ in the abstract, and as ‘When the study population is young children who cannot excrete autonomously, Method C is a practical strategy for sampling certain fecal bacteria of which the difference in abundance is within the LOA.’ in the conclusion section. Based on your insightful comment, we have added the sentence ‘In such studies, the interpretation of the data should be made with caution for bacteria of which the accuracy is reduced’ in the conclusion section (L399-400). Although it is impossible to compare Method A and C in the same children population anymore because the present study is a part of a cohort study, we believe that the results in Research II are worth reporting.

Reviewer 2 Report

The authors clearly describe their study, the results of which may be useful in designing further studies related to the study of stool in children. The limitations of the study are also properly indicated and discussed by the authors. In the discussion, it would be useful to describe more potential applications of the results obtained

Author Response

Dear Reviewer 2

Thank you for giving us the suggestion that is helpful to improve the quality of our manuscript.

We have added the possible application of the tested methods to the elderly in the discussion section (L 381-382).

Round 2

Reviewer 1 Report

I understand that at this point it is impossible to compare toddler samples between method C and method A. However, I believe without that comparison reporting toddlers microbial composition do not add much since there are others manuscripts out there that already reported toddler microbial composition.

Author Response

We sincerely thank for your comment that helped us re-consider appropriateness of our presented data and improve the article quality. In this revision, we have considered the point you have raised again, and revised the manuscript from the following viewpoints.

  • We have omitted the between-age comparison results in Table 5 for some bacteria to assure that all the listed bacteria met our validation criteria that was given in the Method A–C comparison in Research I. The bacteria presented in the revised table are only those of which all the Bland-Altman plots were within the lower or upper limit of the 95% confidence interval of the upper or lower limit, respectively, of standard deviation in Research â… . In other words, the bacteria in the following three categories have been omitted from the original table: (1) bacteria of which the evaluation should not be made based on the results of Research I, (2) bacteria for which whether or not the evaluation can be made is uncertain because the bacteria was not detected in Research I, (3) bacteria for which the evaluation may be possible under the understanding of the reduced accuracy (‘Yes*’ category in the right-most column of the original Table 5). By this revision, I believe that the new Table 5 have met your requirement. The same revision has been made for Figure 8, from which the data of F1 and S10 have been omitted.

  • We have newly cited literatures relating to this study theme in Introduction (Line 49) and in a newly inserted paragraph in the Discussion section (Lines 370-381).

  • We have added more details about the limitation and caution statements in Conclusions.